# Housing European Ground Squirrels (*Spermophilus citellus*) for an Ex Situ Conservation Program

**DOI:** 10.3390/mps7020018

**Published:** 2024-02-20

**Authors:** Boróka Bárdos, Vilmos Altbacker, Henrietta Kinga Török, István Nagy

**Affiliations:** 1Department of Animal Science, Hungarian University of Agriculture and Life Sciences, 40 Guba S., 7400 Kaposvár, Hungary; bardos.boroka@phd.uni-mate.hu; 2Department of Nature Conservation, Hungarian University of Agriculture and Life Sciences, 40 Guba S., 7400 Kaposvár, Hungary; altbacker.vilmos@uni-mate.hu; 3Department of Physiology and Animal Health, Hungarian University of Agriculture and Life Sciences, 40 Guba S., 7400 Kaposvár, Hungary; torok.henrietta.kinga@phd.uni-mate.hu

**Keywords:** European ground squirrel, captive breeding, feed preference, nest material preference

## Abstract

European ground squirrel (*Spermophilus citellus*) populations have declined precipitously over the last 70 years. Its protection cannot be ensured solely by protecting its habitat; it is also necessary to protect the animals ex situ. In our study, within a European ground squirrel species protection program, we examined two elements of indoor housing technology. Knowledge of the animals’ needs is essential for captive housing and breeding success, so in our tests, the animals could freely choose both nest-building materials and feed. In the nest material preference test, the animals could choose from three materials with different structures: paper, Lignocel and hay. In the feed preference test, the animals could also choose from three types of feed: commercial rabbit feed, complete rabbit feed and a natural feed mixture. The first two feeds were in granulated format, and the third was a grain feed mix. Among the nesting materials, they preferred hay, which allowed them to build better-quality nests. Among the feeds, they preferred the grain feed mix, the composition closest to their natural feed, and it was the only one that contained animal protein. Our results contribute to the successful maintenance and breeding the European ground squirrel in captivity.

## 1. Introduction

The European ground squirrel (*Spermophilus citellus*) is an endemic rodent species in Central and Southeastern Europe. Its role is extremely important in the ecosystem of the steppe areas, serving as primary prey for several protected and highly protected birds of prey, such as the eastern imperial eagle (*Aquila heliaca*), the lesser spotted eagle (*Clanga pomarina*) and the saker falcon (*Falco cherrug*). Further, European ground squirrels constitute important prey for other predatory species that contribute to community stability within the Natura 2000 network, such as the steppe polecat (*Mustela eversmanii*) and the marbled polecat (*Vormela peregusna*). The Natura 2000 network created by the European Union is a connected European ecological network that ensures the protection of biological diversity through the protection of natural habitat types and wild animal and plant species of community importance and contributes to maintaining and restoring their favourable nature conservation status [1]. Rodents living underground play an important role in the biotic and abiotic conditions of their habitat. They create new habitats and affect the vegetation (density, spread, and species composition) and the microtopographic characteristics of the soil. They change the soil structure, organic matter and moisture content and affect the amount of biomass [2]. When ground squirrels dig their burrows, they create new habitats, and their digging affects the physical, chemical and biological properties of soils [3], and their burrows help soil ventilation and water management [4]. From the heat, green toads (*Bufo viridis*) and highly protected Hungarian meadow vipers (*Vipera ursinii rakosiensis*) can also crawl into holes dug by ground squirrels [5]. They also play a role in epizoochory [4], which means that animals spread the seeds on their hair.

Since the 1950s, the European ground squirrel has been considered an agricultural pest and was exterminated by large-scale pest control programs. Consequently, we lost nearly 70% of the ground squirrel population in Hungary [6]. Ploughing of grasslands for agricultural production has reduced and fragmented the ground squirrel’s habitat [7]. Ground squirrels are also consumed as a folk food; thus, its population significantly decreased [8]. By the 1980s, the population of ground squirrels had decreased so much [9] that the Ministry of the Environment of Hungary (and other European countries during this period) declared European ground squirrels as protected in 1982, and as in 2001, designated them as highly protected. 

A significant decline was reported in almost all parts of the ground squirrel’s range [9]. The first signs of the decline of the ground squirrel population were already observed in the 1930s in part of a distribution area in the northwest of the Czech Republic [10]. The density of colonies in Northern Serbia was estimated at 30 individuals/hectare at the end of the 1940s, and only 5 individuals/hectare were measured at the end of the 1960s. In Austria, in Burgenland, in the 1950s, ground squirrels were still found in all suitable habitats, and particularly large populations lived in the pastures in the northern and eastern regions of Neusiedlersee/lake Fertő [11]. Detailed data are available from 1968 when 500 ground squirrel holes per hectare were still counted. The Czech Republic can be mentioned as an exception, where the situation of ground squirrels is considered unchanged or better compared to the situation in 2004, which, according to the researchers, is due to the fact that a Species Conservation Action Plan was drawn up in 2008 [10], the implementation of which made habitat management more favourable for ground squirrels [12]. In addition to natural habitat patches, the remaining populations have found refuge in man-made semi-natural short grass grasslands, such as grassy airports [13].

The European Union has developed an action plan for the European ground squirrel [14]. Due to the unfavourable conservation status of the species, several conservation programs were launched to reintroduce the ground squirrel [10,15]. Among the various nature conservation measures, the best and most effective method for the long-term preservation of biological diversity is the protection of habitats, which is possible by preserving living, natural associations and populations; this is called in situ protection. However, in many cases the protection of certain species cannot be solved in their original habitat, or only the protection of the habitat is not sufficient; in such cases protection outside the habitat, i.e., ex situ, is necessary. The European ground squirrel is one species whose protection cannot be solved solely by restoring their original habitat, so protection outside the habitat, ex situ, is necessary. In Hungary, at the beginning of the 2000s, ground squirrel mapping covering the entire country took place [16]. However, the current situation of the ground squirrel in Hungary is not sufficiently clarified, nor is its distribution area known in sufficient detail, which further complicates the protection of the species in its natural habitat. For ex situ protection, knowledge of the species is essential, both from an ecological and ethological point of view, so that we can properly keep and care for them in captivity. The main arenas of ex situ protection are zoos, aquariums, research institutes, breeders, arboretums, and botanical gardens [17]. It should be mentioned that ex situ and in situ conservation are complementary conservation strategies since the ultimate goal of ex situ conservation is to create a viable animal population with adequate genetic diversity that ensures the long-term survival of the species and its reintroduction to natural habitats.

Ex situ protection has improved a lot in recent years; the facilities related to this can perform their tasks much more efficiently, as capture/collection strategies and knowledge of keeping technology have greatly improved [18]. However, it is still ineffective for many less-researched species, such as the European ground squirrel.

For ground squirrels, suitable nest material is crucial for survival and reproduction [19,20,21,22,23,24]. Ground squirrels spend approximately seven months in their underground nest during hibernation [25]. Therefore, the quality of nest material is extremely important for the animals since even the smallest differences in thermal insulation during the long hibernation can have functional significance, as appropriate spring conditions are prerequisite for successful reproduction [25].

Another critical element of keeping ground squirrels in captivity is the acquisition and provision of feed suitable for the species [26], as the active period is relatively short [18], during which time animals must accumulate adequate fat reserves [27,28,29]. The species is considered a herbivore, mainly feeding on green plant parts, flowers, and seeds [10,30]. Its main food plants are variegated crownvetch (*Coronilla varia*), hare’s foot clover (*Trifolium arvense*), field clover (*Trifolium campestre*), zigzag clover (*Trifolium medium*), white clover (*Trifolium repens*), and among the dicotyledons, ribwort plantain (*Plantago lanceolata*), and solidstem burnet saxifrage (*Pimpinella saxifraga*). They consume a small amount of yarrow (*Achillea* sp.) and thyme (*Thymus* sp.) [30]. Furthermore, fescue (*Festuca* sp.), as a food plant, is also an important material for building nests for winter hibernation. Invertebrates, mainly insects, are an essential source of protein, but their diet also includes smaller lizards and bird eggs [31].

Within the framework of the EU LIFE+ project RAPTORSPRAY (2014–2018), the institution operating as Kaposvár University at that time also participated in the breeding program. To promote successful reproduction, the development of husbandry technology is essential. Our first study examined whether the Festuca nesting material used by ground squirrels in nature can be replaced with commercially available nesting materials or whether ground squirrels prefer natural plant nesting material, as was the case in a previous study [32]. In our study, we compare captive ground squirrels’ legally prescribed feed (rabbit feed) by the Department of Environmental Protection and Nature Protection of the Pest County Government Office with a feed mixture closer to the species’ natural feed. We investigated whether ground squirrels preferred commercial diets or a feed closer to their natural diet.

We have little information on the housing and reproduction of European ground squirrels in captivity. The Department of Environment and Nature Protection of the Pest County Government Office determined the recommended housing requirements European ground squirrels in Hungary based on the housing of the American ground squirrel species. Ground squirrels cannot be kept in a smaller space than a 0.43 m × 0. 21 m × 0.20 m clear plastic box, which was originally made for rats. Furthermore, the decree only stipulates that they must be provided with litter. The aim of our preference studies was to improve the husbandry of the captive ground squirrel stock in the laboratory by determining the type of nest material and feed preferred by the animal.

## 2. Materials and Methods

### 2.1. Nest Material Selection Test

The experiment was conducted in the rodent house of the Hungarian University of Agriculture and Life Sciences Kaposvár Campus on the laboratory-born offspring of ground squirrels live-trapped from the wild in different parts of Hungary. The animals were captured and kept based on the registration number PE-KTF/7728-7/2017 of the Department of Environmental Protection and Nature Protection of the Pest County Government Office.

The animals were housed individually in 2-floor Ferplast (Ferplast Favola hamster cage, Castelgomberto, Italy) rodent cages with a floor area of 0.55 m × 0.39 m and a height of 0.28 m at a temperature of 21 °C. In the animal house, the humidity was between 40–60%; in the summer season, the ground squirrels were in the summer room at a temperature of 21 °C with 12 h of daylight (220 lux) and 12 h of dark lighting. During the hibernation period, the animals were housed in 10 lux red lighting at 5 °C. We placed wood shavings at least 5 cm deep on the substrate. Animals had ad libitum access to drinking water and feed. Twenty adult animals participated in the experiment (*n* = 20); ten males and ten females were randomly selected (using an online random number generator). The adult ground squirrels used in the nest material preference study were born in wood shavings. In the experiment, commercially available meadow hay (Bunnynature Freshgrass hay, Bunny Tierernährung GmbH, Melle, Germany), Lignocel (J. Rettenmaier & Söhne GmbH + Co KG, Rosenberg, Germany) and paper nesting material (SAFE Crinklets Natural, SAFE^®^ Complete Care Competence, Rosenberg, Germany) were placed in the hay pockets belonging to the animals’ cages. The hay racks (divided into three parts) were latticed pockets attached to the grid of the cage on the second floor of the cages, in which the three nest-building materials were randomly placed (left, middle or right) using a random sequence generator. 

Based on observations, ground squirrels always build their nests on the floor. Two hundred grams of each material was offered. The duration of the experiment was seven days. The experiment was planned for the summer period so as not to affect reproduction and the subsequent care of the offspring.

The composition of the nests was examined based on the studies of Szenczi et al. [33] and Bilkó et al. [34]. We determined the composition of the materials used for the completed nests by the end of the seventh day by pulling out 20 threads haphazardly from the completed nests and homogenising them. Before sampling nest composition, we also scored the quality of the completed nests based on the studies by Gaskill et al. [35]. The completed nests were given a score of 2 to 5 based on the visual criteria depicted in Figure 1.

The finished nest received 2 points if only a small amount of nest-building material was put together, but no nest shape was formed. The cup shape, where the nest already has a rim, received a score of 3; where this rim is already higher but does not yet close the top, the nest received 4 points, which is a completely closed sphere, received a score of 5 points.

The nesting materials were placed in separate hermetically sealed bags in separate containers. When they were put out for testing, the nest-building materials were measured in separate bags and then placed individually in the cages wearing rubber gloves, avoiding cross-contamination with chemical cues.

### 2.2. Feed Preference Test

The conditions for housing the animals were the same as in the nest material selection test.

Twenty-four adult ground squirrels, 12 males and 12 females participated in the feed preference study (*n* = 24). The test lasted five days. The animals’ access to drinking water was not restricted. We used three different feeds to determine feed preference. The first was a commercially available rabbit feed (Agroszász Ltd., Szászvár, Hungary), the second was Versele-laga Cuni Adult Complete rabbit feed (Versele-Laga GmbH, Deinze, Belgium), and the third was Versele-laga Nature Cuni rabbit feed (Versele-Laga GmbH, Deinze, Belgium). Prior to testing, ground squirrels were provided with the commercially available rabbit feed. The characteristics of the feeds are listed in Table 1 and Table 2. The ground squirrels received the three experimental feeds during the study at 8:00 a.m. In each cage, the three feeds were placed in a bowl (the bowl was divided into three parts). Thirty grams of all three feeds were measured. The weight of the animals was measured at the beginning of the study and also at the end of the study. With the help of video camera recordings, we observed at which of the three feeds the animal went to first. Then, every day after 4 h, we weighed the remaining feed and calculated how much of each feed the animal had consumed. We measured the remaining feed using a digital scale (Vevor Analytical Balance 5000 g × 0.01 g), precise to 0.01 g. The feeds were placed in separate hermetically sealed bags in separate containers. When they were put out for testing, the feeds were measured in separate bags and then placed individually in the cages wearing rubber gloves. Avoiding cross-contamination with chemical cues.

Hereafter, the feeds are denoted by the following abbreviations:○Commercially available rabbit feed (Agroszász Ltd.) = Rabbit feed;○Versele-laga Cuni Adult Complete rabbit feed = Complete;○Versele-laga Nature Cuni rabbit feed = Nature.

#### Statistical Analysis

The data were analysed with the IBM SPSS Statistics 27.0 program package. The Chi-square goodness of fit test was used for the proportion of nest material and the first approach to feed selection. The association between the nest composition (percentages of the paper, Lignocel, and hay) and the nest quality was examined using polyserial correlation. The difference between the sexes in the choice of nest material was using One-Way ANOVA. Repeated Measures ANOVA was used to investigate the feed preference to reveal possible differences between the choice of different types of feed. The Kolmogorov–Smirnov test was used to test normality, a Levene’s test was used to check the equality of standard deviations (the data corresponded to the parametric assumptions).

Results were considered significant where *p* ≤ 0.05.

## 3. Results

### 3.1. Result of Selection of Nest Material

Based on the Chi-square test, we found a significant difference between the three nest materials (*p* = 0.001), with a pronounced preference for hay over other nest-material types, and for paper over Lignocel (Figure 2), although results showed large individual variation.

No significant difference was found between the sexes in the choice of nest-material, neither hay (*p* = 0.432), paper (*p* = 0.405), nor Lignocel (*p* = 1.000).

Looking at the correlation between the nest material composition of the completed nests and the quality of the nest, we found that the amount of paper (*p* = 0.012) and hay (*p* = 0.032) affects the quality of the nest, while Lignocel did not (*p* = 0.261). Furthermore, a larger amount of paper tended to worsen nest quality (r = −0.252), while the presence of hay improved it (r = 0.421).

### 3.2. Results of a Feed Preference Test

We detected a significant difference (*p* = 0.021) in terms of which food ground squirrels approached first, with an apparent preference for the Nature diet over the other two feed types (Figure 3).

Over the 5-day feed preference test, we found a significant difference in the amount of taken feed types (*F*(1, 18) = 2.21 *p* = 0.032). Animals taken the largest percentage of Nature feed (Figure 4).

During the 5-day feed selection, we found a significant difference (*F*(1, 19) = 4.66 *p* = 0.044) in the amount of Nature feed taken between the test days; a slight increase was observed over the 5 days (Figure 4).

The amount of taken Complete feed did not change significantly across the 5 days (Figure 4) (*F*(1, 19) = 0.27 *p* = 0.608). We found a significant difference in the consumption of Rabbit feed among the test days (*F*(1, 19) = 12.21 *p* = 0.002); the amount of taken of Rabbit feed decreased over the course of the experiment (Figure 4).

No significant difference was found between the sexes in the choice of feed (*F*(1, 18) = 0.17 *p* = 0.952).

## 4. Discussion

The result of the choice of nest material is the same as the studies of Gedeon et al. [32], where European ground squirrels studied in the laboratory chose fescue for nest construction; this is the same as our studies, where ground squirrels preferred hay over paper and Lignocel.

Our tests showed that ground squirrels could build better nests using hay. The relationship between better nest shape and thermal insulation capacity was also demonstrated in *Microtus agrestis* nests [36]. The thermal insulation capacity of the nest depends largely on the structure of the nest-building material [36,37]. Materials with a fibrous structure proved to be better for creating the appropriate nest shape for rabbits [34] and mice [38], where the animals preferred materials used in nature. After hay, paper nesting material was preferred by ground squirrels over Lignocel. The paper also has a fibrous structure, which makes it easier for ground squirrels to form the appropriate nest shape. The least preferred material, Lignocel, does not have a fibrous structure but rather is a fine-grained material. The animals probably avoided it because it was unsuitable for creating a nest. In one study, nesting material consisting of paper strips was compared with cotton or cotton wool and found that mice preferred paper strips over other non-fibrous nesting materials [39].

The main function of ground squirrel nests during hibernation is to save energy [19,20,21,22,23,36] which thereby affects survival, as well as subsequent reproductive success [24] since body weight after hibernation is closely related to reproductive success [25].

A suitable quality nest is also very important during the active period of ground squirrels, as the time spent feeding is reduced during the reproductive period [25], so an adequately insulated nest can reduce body weight loss, which increases the survival of the individual and their offspring.

Based on the feed preference test results, captive ground squirrels preferred non-granulated feed. The ground squirrels chose mixture feed (Nature), which contained seeds, vegetables, fruits and animal protein. In terms of composition, the selected feed was the closest to ground squirrels’ feed in nature. These results agree with our previous study, where we examined the feed preference of wild mouse species in laboratory housing, and found that mice, like ground squirrels, preferred the grain mixture feed closest to their natural feed [40]. This can be explained by the fact that the feed mixture consisting of seeds, fruits and vegetables is more similar to the food the ground squirrels eat in nature.

Our feed preference findings are similar to those of Merriman et al. [41], where thirteen-lined ground squirrels (*Ictidomys tridecemlineatus*), native to America, were successfully kept and propagated indoors using a grain feed mixture consisting of seeds, vegetables, and fruits. Furthermore, Merriman et al. [41] described that an important element of the feeding of ground squirrels maintained in indoor housing is the animal protein, which the animals received in the form of mealworms; this is the same as the results of our preference study, where the animals preferred the feed that contained animal protein (Table 1).

The importance of animal proteins is also reported in a study on thirteen-lined ground squirrels [42], according to which the feed intended for rats did not satisfy the needs of the ground squirrels, as they are omnivorous and need animal protein, and hence had their diet supplemented with high-protein cat feed. Vaughan et al. [42] showed that animal protein significantly reduced maternal cannibalism and mortality during hibernation. According to feed preference tests with Belding’s ground squirrels (*Urocitellus beldingi*) [43], the animals preferred to feed with a higher protein or water content, which partially agrees with our results, where the ground squirrels also preferred feed with a higher water content and the second highest protein content. Protein is important in animal growth and development [44]. The insects eaten by ground squirrels in nature are rich in protein (30–68% dry matter) and have significant amino acid content. Daurian ground squirrels (*Spermophilus dauricus*), native to Central Asia, have been observed to consume larger quantities of insects and plant seeds during the spring breeding season and the autumn pre-hibernation season [45]. Adequate feeding during the reproductive period affects the survival of the offspring [46], so this period is particularly important for higher fat and protein supplementation.

Increased protein and fat intake in the autumn period is important to achieve proper conditions so that the animals do not die during the long winter hibernation [47]. For hibernating small mammals, the fat content of the feed is very important, as it can influence hibernation success [48,49]. The nutrient intake of ground squirrels increases dramatically two months before the start of hibernation, and they reach a body fat content of 35–40% before hibernation [50,51].

According to some research, feeds with a higher fatty acid content consumed during this period promote better hibernation [52]. For example, in laboratory experiments, the golden-mantled ground squirrel (*Callospermophilus lateralis*) preferred feed with a higher fatty acid content in feed preference tests [52]; similar to our feed preference results, where ground squirrels also preferred feed with a higher fat content. In addition to proteins, the role of polyunsaturated fatty acids is also important. The right fat composition helps to optimize hibernation and can increase the length of torpor phases, which is more favourable for the animal in terms of energy consumption [53]. The torpor phase represents the resting phase of hibernation, in which case the metabolism slows down [54].

In summary, ground squirrels housed indoors, like their congeners living in the wild, chose grasses, i.e., hay, as a nest-building material among the offered materials. The hay contributed to forming a better-quality nest; thus, promoting heat retention and successful overwintering, increasing reproductive success after emergence from hibernation.

Based on the feed preference test results, ground squirrels kept and bred indoors chose the feed mixture closest to their natural diet, the fat and animal protein content of which can contribute to successful hibernation and subsequent reproductive success.

## 5. Conclusions

The main task of ex situ protection is to ensure successful reproduction of the species in captivity; thus, finding suitable nest-building material and feed can be extremely important in promoting successful reproduction.

In the study, the captive ground squirrels, like their wild counterparts, prefer hay as nest-building material. One of the basic conditions of a successful species conservation program is that the housing technology of animals kept and bred in closed spaces is adapted to the natural needs of the species. In the case of ground squirrels kept in cages, the nesting material is extremely important, as they cannot dig holes in the cage, which protects them from external environmental factors. Hay, as a nest-building material, has a long, fibrous structure from which ground squirrels can build a nest of adequate quality, which insulates it during hibernation, thus protecting the animal from unnecessary heat loss and thus from losing body weight.

The feed preference test revealed that the ground squirrels preferred the seed mixture containing fruits, vegetables and animal protein, which are closest to their natural diet, over the granulated rabbit feed. According to the literature, ground squirrels in nature feed on green parts of plants, fruits, insects, and seeds. In the case of hibernating small mammals, proper feeding is extremely important; the active period of ground squirrels is very short, and only a few months are available to accrue adequate fat reserves. An adequate fat reserve is necessary for hibernation and for successful reproduction upon emergence from hibernation. Males that emerge with a higher body weight have better chances of controlling a territory and outcompeting other males for access to receptive females [55]. Females that emerge from hibernation with a higher body mass enjoy greater reproductive success in terms of becoming impregnated [46,48] and rearing offspring to weaning [46].

Our results contribute to successful husbandry and captive-breeding aimed at preserving the species by developing appropriate housing technology. The successful breeding program contributes to the return of the species to its natural environment.

## Figures and Tables

**Figure 1 mps-07-00018-f001:**
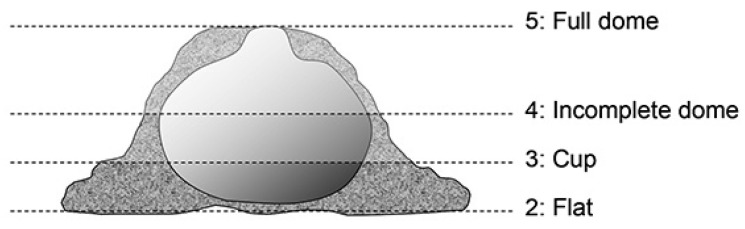
Diagrammatic representation of nest quality scoring.

**Figure 2 mps-07-00018-f002:**
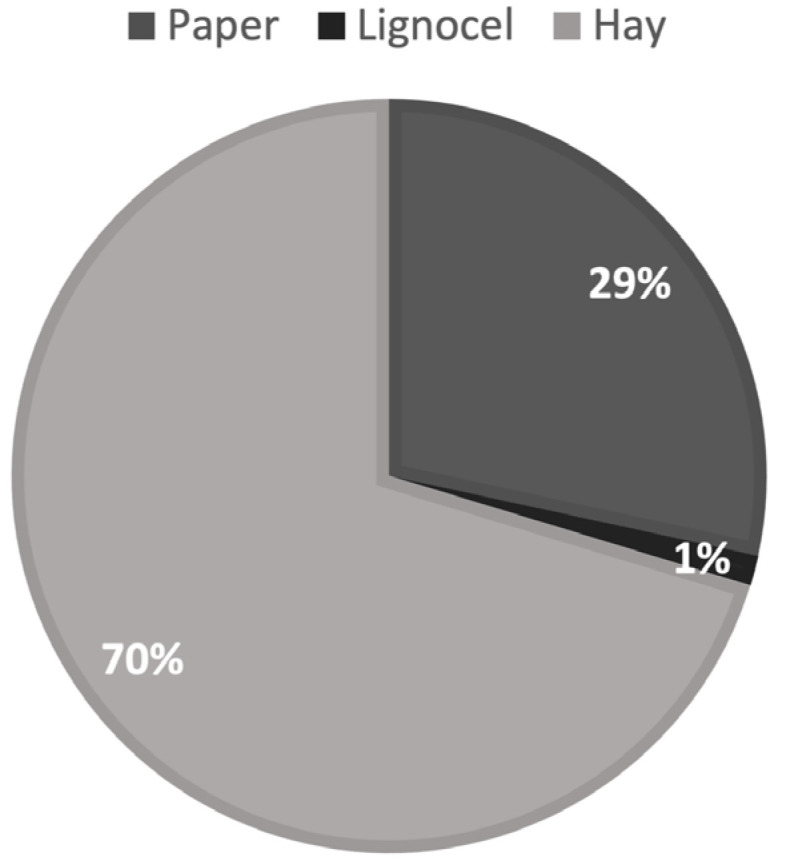
Nest material preference of ground squirrels.

**Figure 3 mps-07-00018-f003:**
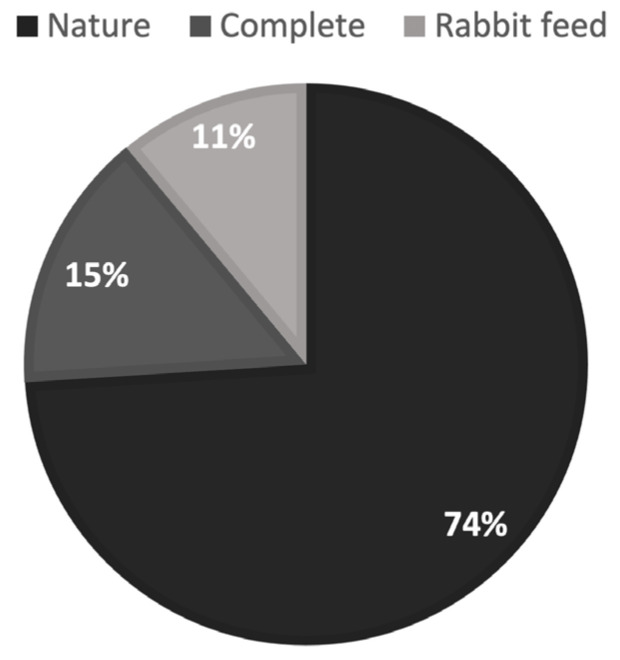
Ground squirrels’ first choice of feed.

**Figure 4 mps-07-00018-f004:**
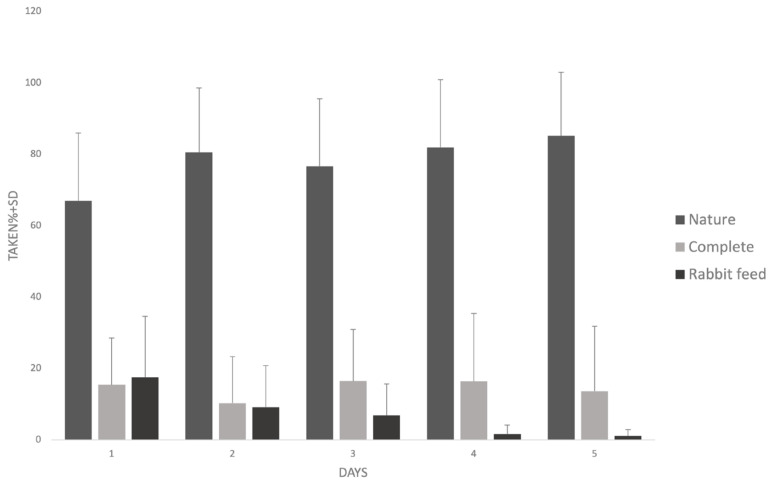
The percentage of feeds taken in 5 days.

**Table 1 mps-07-00018-t001:** Composition of the feeds used in the preference test.

Feed Name	Feed Structure and Diameter	Components
Commercially available rabbit feed (Agroszász Ltd.)	granulated (3.3 mm)	grass pellets,
		lucerne,
		extracted sunflower semolina, carrot slices,
		wheat,
		wheat bran,
		oats,
		barley,
		full-fat soy,
		additives.
Versele-laga Cuni Adult Complete rabbit feed	granulated (5.4 mm)	grass pellets,
		carrot,
		vegetable protein extracts,
		linseed,
		fructo-oligosaccharides,
		marigold,
		yucca.
Versele-laga Nature Cuni rabbit feed	mixture	grass pellets,
		green peas,
		carrot,
		parsnip,
		apple,
		animal protein,
		oils and fats,
		fructo-oligosaccharides,
		mannan oligosaccharides,
		marigold,
		chlorella algae,
		yucca.

**Table 2 mps-07-00018-t002:** Nutrient composition of the feeds (dry matter basis).

Nutrient	Commercially Available Rabbit Feed (Agroszász Ltd.)	Versele-Laga Cuni Adult Complete Rabbit Feed	Versele-Laga Nature Cuni Rabbit Feed
Crude protein (%)	16	14	14.7
Crude fat (%)	3.3	3	3.5
Crude fibre (%)	15	20	16.5
Crude ash (%)	7	7.5	8.3
Moisture content (%)	7.4	7.5	7.6

## Data Availability

After acceptance of the manuscript, data will be archived in the institutional repository of the MATE University Kaposvár Campus.

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
