# Peer review of "Housing European Ground Squirrels (Spermophilus citellus) for an Ex Situ Conservation Program"

_mps, 2024, doi:10.3390/mps7020018_

Round 1
Reviewer 1 Report
Comments and Suggestions for Authors
The manuscript presents the results of controlled laboratory experiments assessing nest material and food preferences of captive European ground squirrels. Given the need to bolster wild populations with captive-bred individuals to ensure the persistence of this keystone species, the study itself is worthwhile, and appears to have been conducted appropriately, providing robust results that inform those who maintain captive-bred squirrels for conservation purposes, or basic research. I perceive no fundamental flaws in the design or execution of the study. The Introduction provides adequate background information establishing clear rationale for the study (but see comments 3-6 below). The Methods are, for the most part, comprehensive and clearly presented (but see comments 7-17 below), as are the Results (but see comments 18-21). The Discussion highlights the main findings and the implications of those and is generally well written, though a more conservative attribution of the factors promoting the documented feed preference is required (see comments 22-27). I have made suggestions for minor typographical and grammatical revisions on the attached annotated PDF, yet beyond those, recommend the authors attend to the following concerns:
1) The title of the manuscript can be simplified considerably yet still reflect the content of the manuscript.
2) The opening sentence of the abstract should be modified to provide the reader with some idea of the time period over which population declines have been noted, and the three separate clauses conjoined by semi-colons in the present draft should be divided into two complete sentences.
3) The opening paragraph of the Introduction highlights the importance of European ground squirrels as a prey base for a variety of avian and terrestrial predators, but does their ecological importance go beyond that in modifying plant and/or insect communities, or as disease vectors?
4) On lines 43-44 of the Introduction, the agency that designated this ground squirrel species as protected in 1982, and highly protected in 2001 should be reported.
5) On line 74, the authors refer to "legally defined" feed for European ground squirrels, but do not report the organization or agency legally prescribing or recommending that feed, and thus need to report that.
6) In the final paragraph of their Introduction, the authors state that there is little information on maintaining European ground squirrels in captivity, which suggests there is some, and thus, it is essential that the authors summarize what is already known here.
7) In reporting the physical factors squirrels were maintained and tested under, the authors should also report relative humidity and the lighting regime (source, intensity and photoperiod).
8) Line 89: What is a 2-floor Ferplast rodent cage? Can manufacturer information be provided so that readers unfamiliar with this cage type can identify it?
9) On line 93, the authors report random selection of subjects, random placement of nest material piles on line 95, random sampling of fibres from nests on line 103, random assignment of subject to feed conditions on line 113 but were these really randomized or arbitrary? If truly random, the method of randomization should be reported. Further, the authors must clarify what they mean by placing the nest materials in the hay pockets of animals' cages. What are hay pockets and were these evenly dispersed and equidistant from the nesting area? Which, if any, of the three nest materials were young born into in captivity?
10) Figure 1 (lines 106-107) provides a useful diagrammatic representation of how nest quality was scored, but requires and informative caption and must be referred to by its Figure number in the body of the manuscript (i.e. Figure 1) instead of "on the figure below" on line 105.
11) Lines 112-113: how many of the subjects employed in the food preference test were male and how many were female? Further, which of the three feeds employed in the food preference experiment, if any, had squirrels been maintained on prior to testing?
12) On line 119, the authors report 3 feed bowls offering 30 g of each of three feed types being offered within each cage to individually-housed subjects, but must report how the feed-containing bowls were positioned. Further, in that the description of the method here implies that the authors appropriately conducted a simultaneous-choice test allowing each subject free choice of the three feed types, this makes me wonder why they talk about dividing the 24 subjects in three groups of 8 individuals on lines 112-113 above.
13) On lines 120-121 the authors must report how animals were weighed.
14) On line 124, the authors appear to assume that any food missing from the bowls was consumed by subjects, but is there any possibility that food removed from bowls was remained uneaten (i.e. was cached or simply displaced from the bowls)?
15) In describing the methods of analysis, it is unclear how the authors employed a chi-square test to examine proportionate nest composition as the chi-square test requires meristic data. I'm guessing they used a goodness-of-fit test to detect departure from a null expectation of equivalent numbers of threads of the three material types, but they must report the structure of their test. Further, I am wondering if data were simply pooled across replicates, as this seems to be a case where one could use a heterogeneity chi-square test to meaningfully test for significant variation in nest material preferences among subjects. Further, it is unclear how the authors contrasted male versus female preferences. Similarly, the structure of the repeated measures ANOVA the authors report as having been employed to analyse their food preference data needs to be reported, along with whether or not the data for the weight of food of the three types taken from the dishes satisfied the parametric assumptions of normality and homoscedasticity. Finally, it is unclear what statistical test was employed to determine whether the food dish subjects approached first was dependent upon food type.
16) The authors should report the probability of Type I error at which they deemed the results of their tests statistically significant. (i.e. add a final sentence the final paragraph in the methods stating "Results were considered significant where P ≤ 0.05."
17) The authors should report whether the nest materials and feeds offered to squirrels in their experimental tests were handled by hand or using distinct utensils to avoid cross-contamination with chemical cues.
18) At present the pie graphs showing the proportionate use of the three nest material types (Figure 2) and the food type first approached (Figure 3) are entirely redundant with the percentage values presented in the text of the manuscript. Use one or the other, but not both. If you do keep the figures instead of reporting the percentages in the text of your results, then be sure to refer the reader to figures by figure number in the text.
19) In reporting the results of your statistical tests (lines 145, 152, 153, 158), provide the exact P value rather than a range relative to a fixed probability.
20) Mention of the subject's previous familiarity with rabbit feed should be mentioned in the Methods (and Discussion) and not in the Results section (line 164) of the manuscript.
21) Be sure to refer to the figure depicting the results you are describing within your Results section (see annotated PDF), use the word "among" rather than "between" where more than two groups are being contrasted, and use the word sexes to refer to males versus females as gender explicitly refers to roles while the biological distinction of male versus female is referred to as one's sex.
22) Delete the opening sentence of the Discussion (lines 181-182) as that goes without saying.
23) While the authors offer a reasonable interpretation as to why squirrels would be selected to prefer hay over other nest materials, I feel they should also at least speculate on why paper was not as actively avoided as lignocel.
24) Based on the data provided in Table 2, all three feeds contained animal protein, and thus the sentence stating that squirrels preferred the feed with animal protein (lines 214-215) should be revised as indicated on the annotated PDF.
25) The Discussion focuses too strongly on the presence of animal protein in the Nature feed accounting for the documented preference of the squirrels for the Nature feed over the other two feed types. While the animal protein difference could constitute the primary basis of the food preference documented, and discission of how this fits with the results of previous studies documenting the importance of animal protein (as the authors have done) is worthwhile, the Discussion should take a more conservative approach, reiterating what preferences were evident, highlighting the multiple differences that may contribute to that preference (e.g. could it have been the presence of apple, green peas or other components in the preferred diet or an aversion to some unique components of the non-preferred diets, e.g. marigold, oats, barley etc., that accounted for the results obtained, or perhaps differences in crude ash or fat content, the latter of which the authors at least get to toward the end of the present Discussion), or even other differences among the feed types offered (texture, granularity, coloration, etc.) and then outlining what corroborating evidence exists for the importance of each of those dietary components or attributes. In essence, where multiple differences exist among feed types, a more cautious interpretation of what may account for food preferences is warranted. The present concluding statement (lines 240-241) attributing the documented feed preference to fat and animal protein exemplifies how the present interpretation goes beyond the scope of what can rightfully be concluded based on the many differences (Tables 1 and 2) among the three feed types.
26) On lines 227-228, the authors state that "the nutrition of ground squirrels increases dramatically two months before the start of hibernation", though by "nutrition" are they referring to foraging rate or nutrient intake?
27) On line 231, what is meant by "better hibernation"?
I hope my comments prove useful to the authors, and look forward to reading the revised manuscript. I wish all those involved in this study every continued happiness and success in their ongoing research pursuits.
Sincerely yours,

The paper is generally well-written, though I have attached an annotated copy of the PDF with numerous suggestions for grammatical revisions that will enhance the clarity of the manuscript.
Author Response
We thank the reviewer for the thorough work and the attached PDF. The suggestions and remarks helped us improve the manuscript.
1) The title of the manuscript can be simplified considerably yet still reflect the content of the manuscript.
Title simplified according to the reviewer's suggestion
2) The opening sentence of the abstract should be modified to provide the reader with some idea of the time period over which population declines have been noted, and the three separate clauses conjoined by semi-colons in the present draft should be divided into two complete sentences.
Corrected and the abstract was supplemented with the time period.
3) The opening paragraph of the Introduction highlights the importance of European ground squirrels as a prey base for a variety of avian and terrestrial predators, but does their ecological importance go beyond that in modifying plant and/or insect communities, or as disease vectors?
Thank you for the suggestion, we have added some information to the text.
4) On lines 43-44 of the Introduction, the agency that designated this ground squirrel species as protected in 1982, and highly protected in 2001 should be reported.
Added to the text
5) On line 74, the authors refer to "legally defined" feed for European ground squirrels, but do not report the organization or agency legally prescribing or recommending that feed, and thus need to report that.
Added to the text.
6) In the final paragraph of their Introduction, the authors state that there is little information on maintaining European ground squirrels in captivity, which suggests there is some, and thus, it is essential that the authors summarize what is already known here.
Thank you for your comment, we have added the requested information to the text.
7) In reporting the physical factors squirrels were maintained and tested under, the authors should also report relative humidity and the lighting regime (source, intensity and photoperiod).
We have added the requested information to the text.
8) Line 89: What is a 2-floor Ferplast rodent cage? Can manufacturer information be provided so that readers unfamiliar with this cage type can identify it?
Added to the text.
9) On line 93, the authors report random selection of subjects, random placement of nest material piles on line 95, random sampling of fibres from nests on line 103, random assignment of subject to feed conditions on line 113 but were these really randomized or arbitrary? If truly random, the method of randomization should be reported. Further, the authors must clarify what they mean by placing the nest materials in the hay pockets of animals' cages. What are hay pockets and were these evenly dispersed and equidistant from the nesting area? Which, if any, of the three nest materials were young born into in captivity?
A numerical code identifies ground squirrels. The ground squirrels participating in the study were selected using a random number generator based on the numerical code ( https://kiszamolo.com/veletlenszam-generator/ ).
We have supplemented the section on the hay pocket with some information for better understanding.
The ground squirrels were kept in deep litter (wood shavings) before the test.
10) Figure 1 (lines 106-107) provides a useful diagrammatic representation of how nest quality was scored, but requires and informative caption and must be referred to by its Figure number in the body of the manuscript (i.e. Figure 1) instead of "on the figure below" on line 105.
Thank you for your comment, we have added the text.
11) Lines 112-113: how many of the subjects employed in the food preference test were male and how many were female? Further, which of the three feeds employed in the food preference experiment, if any, had squirrels been maintained on prior to testing?
Added the text. In the nest material preference test, the ratio of the sexes was 50-50%; the animals had already met commercially available rabbit feed before the test.
12) On line 119, the authors report 3 feed bowls offering 30 g of each of three feed types being offered within each cage to individually-housed subjects, but must report how the feed-containing bowls were positioned. Further, in that the description of the method here implies that the authors appropriately conducted a simultaneous-choice test allowing each subject free choice of the three feed types, this makes me wonder why they talk about dividing the 24 subjects in three groups of 8 individuals on lines 112-113 above.
Corrected in text. A description of another study was mixed into the text.
13) On lines 120-121 the authors must report how animals were weighed.
The digital hundredth-gram scale added the text.
14) On line 124, the authors appear to assume that any food missing from the bowls was consumed by subjects, but is there any possibility that food removed from bowls was remained uneaten (i.e. was cached or simply displaced from the bowls)?
There is a chance that the ground squirrels stored the feed, but based on the observation, they only store the feed they like, so the results are not influenced by which feed the ground squirrels preferred.
15) In describing the methods of analysis, it is unclear how the authors employed a chi-square test to examine proportionate nest composition as the chi-square test requires meristic data. I'm guessing they used a goodness-of-fit test to detect departure from a null expectation of equivalent numbers of threads of the three material types, but they must report the structure of their test. Further, I am wondering if data were simply pooled across replicates, as this seems to be a case where one could use a heterogeneity chi-square test to meaningfully test for significant variation in nest material preferences among subjects. Further, it is unclear how the authors contrasted male versus female preferences. Similarly, the structure of the repeated measures ANOVA the authors report as having been employed to analyse their food preference data needs to be reported, along with whether or not the data for the weight of food of the three types taken from the dishes satisfied the parametric assumptions of normality and homoscedasticity. Finally, it is unclear what statistical test was employed to determine whether the food dish subjects approached first was dependent upon food type.
We confirm that the goodness abd-fit thest was used in a way as supposed by the reviewer because our data was in fact countable (20 samples were taken from each nest). We note that we had not intention to test if the various individuals showed different preferences. The Kolmogorov–Smirnov test was used to test normality, a Levene's test was used to check the equality of standard deviations. The first feed approach was evaluated also by means of chi-square test similarly to that of nest material preference.
16) The authors should report the probability of Type I error at which they deemed the results of their tests statistically significant. (i.e. add a final sentence the final paragraph in the methods stating "Results were considered significant where P ≤ 0.05."
Added the text.
17) The authors should report whether the nest materials and feeds offered to squirrels in their experimental tests were handled by hand or using distinct utensils to avoid cross-contamination with chemical cues.
Feed and nesting materials were also in separate hermetically sealed bags, placed in separate containers. When placing them for the test, the feed and nest-building materials were measured in separate bags, and then they were placed individually in the cages wearing rubber gloves.
18) At present the pie graphs showing the proportionate use of the three nest material types (Figure 2) and the food type first approached (Figure 3) are entirely redundant with the percentage values presented in the text of the manuscript. Use one or the other, but not both. If you do keep the figures instead of reporting the percentages in the text of your results, then be sure to refer the reader to figures by figure number in the text.
Corrected in text.
19) In reporting the results of your statistical tests (lines 145, 152, 153, 158), provide the exact P value rather than a range relative to a fixed probability.
Corrected in text.
20) Mention of the subject's previous familiarity with rabbit feed should be mentioned in the Methods (and Discussion) and not in the Results section (line 164) of the manuscript.
Corrected in text.
21) Be sure to refer to the figure depicting the results you are describing within your Results section (see annotated PDF), use the word "among" rather than "between" where more than two groups are being contrasted, and use the word sexes to refer to males versus females as gender explicitly refers to roles while the biological distinction of male versus female is referred to as one's sex.
Corrected based on PDF.
22) Delete the opening sentence of the Discussion (lines 181-182) as that goes without saying.
Corrected based on PDF.
23) While the authors offer a reasonable interpretation as to why squirrels would be selected to prefer hay over other nest materials, I feel they should also at least speculate on why paper was not as actively avoided as lignocel.
Thank you for the suggestion, we have supplemented the given section in the text.
24) Based on the data provided in Table 2, all three feeds contained animal protein, and thus the sentence stating that squirrels preferred the feed with animal protein (lines 214-215) should be revised as indicated on the annotated PDF.
Corrected based on PDF.
25) The Discussion focuses too strongly on the presence of animal protein in the Nature feed accounting for the documented preference of the squirrels for the Nature feed over the other two feed types. While the animal protein difference could constitute the primary basis of the food preference documented, and discission of how this fits with the results of previous studies documenting the importance of animal protein (as the authors have done) is worthwhile, the Discussion should take a more conservative approach, reiterating what preferences were evident, highlighting the multiple differences that may contribute to that preference (e.g. could it have been the presence of apple, green peas or other components in the preferred diet or an aversion to some unique components of the non-preferred diets, e.g. marigold, oats, barley etc., that accounted for the results obtained, or perhaps differences in crude ash or fat content, the latter of which the authors at least get to toward the end of the present Discussion), or even other differences among the feed types offered (texture, granularity, coloration, etc.) and then outlining what corroborating evidence exists for the importance of each of those dietary components or attributes. In essence, where multiple differences exist among feed types, a more cautious interpretation of what may account for food preferences is warranted. The present concluding statement (lines 240-241) attributing the documented feed preference to fat and animal protein exemplifies how the present interpretation goes beyond the scope of what can rightfully be concluded based on the many differences (Tables 1 and 2) among the three feed types.
Unfortunately, the feed tests with ground squirrels were limited to protein and fat composition, so we could mainly compare our results with the results of these studies. Nevertheless, we tried to improve and extend the discussion also from other aspects.
26) On lines 227-228, the authors state that "the nutrition of ground squirrels increases dramatically two months before the start of hibernation", though by "nutrition" are they referring to foraging rate or nutrient intake?
Corrected text for nutrient intake.
27) On line 231, what is meant by "better hibernation"?
The section in the text has been supplemented, giving additional information.
Reviewer 2 Report
Comments and Suggestions for Authors
GENERAL COMMENT:
I consider this work is within the scope of “Methods and Protocols”. It contains information useful in a field in which available information is very scarce and of relevance for European squirrel populations’ recovery. Overall, it is well organised.
I indicate below specific points to be improved in the manuscript.
TITLE:
I suggest to modifying as follows: “…participating in an ex-situ conservation program”, rather than “…participating in the ex-situ conservation program”. However you can leave your original title if you are completely sure that the one described in the manuscript is the only ex-situ conservation program in the world.
ABSTRACT:
Overall, it is OK.
In the same vein as for the title, in Line 15 you can say “within the European ground squirrel species protection program” only if you are completely sure that the one described in the manuscript is the only ex-situ conservation program in the world. If not, it is preferably to write: “within a European ground squirrel species protection program”.
Line 20: Insert “:” at “three types of feed commercial rabbit feed, complete rabbit feed and”, thus resulting as “three types of feed: commercial rabbit feed, complete rabbit feed and”.
Lines 21-22: Please rewrite the following sentence in past tense: “The first two feeds are in granulated format, and the third is a grain feed 21 mix.”
KEYWORDS:
These are OK.
INTRODUCTION:
Lines 30-37: this paragraph need some bibliographic citations to support the information contained in it (for example, many international readers do not known what is Natura 2000).
Line 42: Please revise whether “herd” is a proper word to refer to the overall population of a wild species.
Line 56: Please check whether it is better “essential for the success of reproduction” than “essential for the survival of reproduction”.
Line 60: Are you sure that “calving” is the proper term for giving birth squirrels? It is usually used for cows. If there is a more proper term I suggest using it.
By the end of the Introduction section: please formulate the aim of the study in the classical way proper of scientific publications: “This study aimed to…”
MATERIALS AND METHODS:
Overall, this section is well organised.
Line 91: Type “ad libitum” in italics.
Line 96: Material and methods must be written in past tense: “each material was”, rather than “each material is”
Line 110: “keeping the animals were the same”, rather than “keeping the animals are the same”
Subsection “2.2. Feed preference test”. An adaptation period to the experimental feed before the feed intake recording was stablished? If yes, please indicate it.
RESULTS SECTION:
Overall, this section is OK.
Lines 149-155: Please write in past tense (for example, hay affected the quality of the nest…, chose rather than choose).
DISCUSSION SECTION:
Overall, this section is OK.
CONCLUSIONS:
This section must be improved by specifically indicating the better alternative for feed and for nesting material, derived from your results. The current version is unspecific.
CITATION OF BIBLIOGRAPHIC REFERENCES IN THE TEXT:
Please adjust the citations in explicit form to the style of the journal. For example, if you cite “Szenczi et al. (2011) [25]”, the correct way is: “Szenczi et al. [25]”. Revise this flaw in Lines 101, 105, 183, 208, 211, 219…
REFERENCES SECTION:
In general terms, this section has a good adjustment to the style and format of the journal for references. However, I recommend reviewing it for removing typos and correct potential flaws. For example:
Latin names of the organisms must be typed in italics.
Etc.
TABLES:
Table 2: Do not put “%” to each number. Put it with the variable name: “Crude protein (%)”.
Table 2: Please indicate whether values refer to as-fed basis or on dry matter basis.
FIGURES:
Circles in figures 2 and 3 are too big.
Author Response
We thank the reviewer the corrections and remarks. Our detailed responses are as follows:
TITLE:
I suggest to modifying as follows: “…participating in an ex-situ conservation program”, rather than “…participating in the ex-situ conservation program”. However you can leave your original title if you are completely sure that the one described in the manuscript is the only ex-situ conservation program in the world.
Manuscript title corrected.
ABSTRACT:
Overall, it is OK.
In the same vein as for the title, in Line 15 you can say “within the European ground squirrel species protection program” only if you are completely sure that the one described in the manuscript is the only ex-situ conservation program in the world. If not, it is preferably to write: “within a European ground squirrel species protection program”.
Corrected in text.
Line 20: Insert “:” at “three types of feed commercial rabbit feed, complete rabbit feed and”, thus resulting as “three types of feed: commercial rabbit feed, complete rabbit feed and”.
Corrected in text.
Lines 21-22: Please rewrite the following sentence in past tense: “The first two feeds are in granulated format, and the third is a grain feed 21 mix.”
Corrected in text.
KEYWORDS:
These are OK.
INTRODUCTION:
Lines 30-37: this paragraph need some bibliographic citations to support the information contained in it (for example, many international readers do not known what is Natura 2000).
Added to the text.
Line 42: Please revise whether “herd” is a proper word to refer to the overall population of a wild species.
Corrected in text.
Line 56: Please check whether it is better “essential for the success of reproduction” than “essential for the survival of reproduction”.
Corrected in text.
Line 60: Are you sure that “calving” is the proper term for giving birth squirrels? It is usually used for cows. If there is a more proper term I suggest using it.
Corrected in text.
By the end of the Introduction section: please formulate the aim of the study in the classical way proper of scientific publications: “This study aimed to…”
Corrected in text.
MATERIALS AND METHODS:
Overall, this section is well organised.
Line 91: Type “ad libitum” in italics.
Corrected.
Line 96: Material and methods must be written in past tense: “each material was”, rather than “each material is”
Corrected.
Line 110: “keeping the animals were the same”, rather than “keeping the animals are the same”
Corrected.
Subsection “2.2. Feed preference test”. An adaptation period to the experimental feed before the feed intake recording was stablished? If yes, please indicate it.
The animals were encountered with commercially available rabbit feed before the test (supplemented in the text). The other feeds were encountered on the first day of the test.
RESULTS SECTION:
Overall, this section is OK.
Lines 149-155: Please write in past tense (for example, hay affected the quality of the nest…, chose rather than choose).
Corrected in text.
DISCUSSION SECTION:
Overall, this section is OK.
CONCLUSIONS:
This section must be improved by specifically indicating the better alternative for feed and for nesting material, derived from your results. The current version is unspecific.
Supplemented in the text.
CITATION OF BIBLIOGRAPHIC REFERENCES IN THE TEXT:
Please adjust the citations in explicit form to the style of the journal. For example, if you cite “Szenczi et al. (2011) [25]”, the correct way is: “Szenczi et al. [25]”. Revise this flaw in Lines 101, 105, 183, 208, 211, 219…
Corrected in text.
REFERENCES SECTION:
In general terms, this section has a good adjustment to the style and format of the journal for references. However, I recommend reviewing it for removing typos and correct potential flaws. For example:
Latin names of the organisms must be typed in italics.
Etc.
Corrected.
TABLES:
Table 2: Do not put “%” to each number. Put it with the variable name: “Crude protein (%)”.
Corrected in table 2.
Table 2: Please indicate whether values refer to as-fed basis or on dry matter basis.
We sent the samples to an accredited laboratory for analysis. It was a dry matter basis. Added to the text.
FIGURES:
Circles in figures 2 and 3 are too big.
Corrected.
Reviewer 3 Report
Comments and Suggestions for Authors
It should be noted that the authors undertook research on the functioning of the population of a species that was until recently very common and even considered a pest, and is now almost extremely endangered, the ground squirrel. Dealing with this issue should be assessed as very important and timely. However, after reading the manuscript, I have a few comments that I think will improve this already valuable study.
Line 40, why the reference is only to Hungary. I understand that this is the authors' country, but the introduction should provide a general description of the European situation of this species. After all, protection programs are carried out in many countries. Even the authors themselves write about it later in the introduction.
In my opinion, the conclusions should be rewritten because they are too modest and not fully consistent with the research. After all, the authors did not comprehensively evaluate the technologies for keeping ground squirrels, but only selected aspects. In my opinion, the summary and conclusions can form one whole and then it will certainly be better for the work and the potential recipient.
Taking into account the above minor comments, which in my opinion will allow for improvement, the manuscript should be published.
Author Response
Authors thank the reviewer evaluating our manuscript.
Our responses are as follows:
Line 40, why the reference is only to Hungary. I understand that this is the authors' country, but the introduction should provide a general description of the European situation of this species. After all, protection programs are carried out in many countries. Even the authors themselves write about it later in the introduction.
Thank you for the suggestion, we have added the paragraph to the European situation of ground squirrels.
In my opinion, the conclusions should be rewritten because they are too modest and not fully consistent with the research. After all, the authors did not comprehensively evaluate the technologies for keeping ground squirrels, but only selected aspects. In my opinion, the summary and conclusions can form one whole and then it will certainly be better for the work and the potential recipient.
We rewrote the conclusion section to be more consistent with the research findings.
Round 2
Reviewer 1 Report
Comments and Suggestions for Authors
I appreciate the authors taking the comments made on their initial submission to heart, as I feel the revised manuscript is vastly improved. In completing their revision, the authors have adequately dealt with each of the 27 numbered concerns I raised in my initial review, except for previous concern numbers:
9) The authors should report use of online random number generator to select 10 males and 10 females for the preference test. This can be achieved simply by inserting a parenthetical statement "(using an online random number generator)" after reporting "randomly selected" before the period that ends the sentence on Line 154. My concern remains about how nest materials were "randomly" assigned to hay pockets (Line 156), as even in their reply, the authors do not describe how this was randomized, and thus leaves concern that the dispersion of nest material types offered may have influenced the preference of subjects. I am also unclear on what a "latticed pocket" (Line 157) is, and thus suggest this be clarified for readers. Line 165 still reports pulling 20 threads at random from completed nests for analysis, without reporting how randomization was achieved. If no formal method of randomization was employed to select threads, then the authors should report that threads were selected arbitrarily or haphazardly, not randomly. Finally, the authors should explicitly that the adults employed as subjects in the nest material preference study were born into wood shavings (which I infer from their reply to my comment 9 in the previous review) as early experience has a pronounced influence on subsequent preferences.
12) While the authors have removed the erroneous statement about dividing the 24 subjects into three groups of 8, and reported that the three feed types were presented individually in separate bowls, they have not described where the bowls were positioned, and whether placement was at all randomized or systematically controlled across trials to avoid biasing subject preference as a product of bowl placement. I also think the authors would be best served by indicating that subjects were offered a simultaneous choice of the three feed types somewhere in outlining their feed preference experiment.
13) The authors report the use of a digital scale, precise to 0.01 g to weigh feed, but should provide the model number and manufacturer of that scale, and mention in the context of weighing subject ground squirrels at the beginning and end of the study (lines 191-192) whether that same scale was used.
14) In response to my comment, the authors acknowledge that not all food taken from the bowls by subjects was necessarily consumed. Thus the terms "consumed" and "consumption" in describing the results of the feed preference experiment are not technically correct, and should either replaced with "taken" or the authors must define their use of the term "consumed" in their methods, to indicate that by consumed they are referring to food removed from the bowls either through immediate ingestion, or caching for subsequent ingestion by subjects.
15) While the authors have provided information to clarify and alleviate my concerns regarding the statistical analysis of their results in their reply to my comment 15 in my original review, they have not incorporated that into their revised manuscript, and should do so, so as to dispel similar uncertainty on the part of readers who may or may not make appropriate assumptions regarding the same issues I raised. I also remain puzzled as to why the authors do not present, nor contrast, nest material preferences of male and female ground squirrels when they explicitly report testing 10 males and 10 females in their nest-material preference test. Finally, at least some mention of inter-individual variation in preferences would be worthwhile, even if that is not tested for formally using the heterogeneity chi-square test mentioned in my original comment 15.
17) The authors reply to my concern regarding contamination of nest materials and feed offered with unwanted chemical cues alleviates the concern expressed in my original comment 17, but, the authors should report these precautions to readers in describing their methods in their revised manuscript.
18) The authors have eliminated the redundancy between their textual account and the Figures incorporated within their results in their revised manuscript, but still must explicitly refer readers to Figures 3 and 4 (see suggested revisions in the annotated PDF to achieve this economically for Figures 3 and 4 in the Results section).
25) While the authors have done a much better job of keeping their inferences to the scope their study allows, there should still be an additional paragraph incorporated toward the end of the Discussion, acknowledging that the extent to which other attributes that varied among the feed types offered (e.g. the presence or absence of specific constituents described in column 3 of Table 1) or nutritional properties like crude ash (Table 2) contribute to the documented preference.
I also suggest a few additional minor corrections, including certain grammatical revisions marked directly on the annotated PDF returned with my review, and:
1) Revising the wording of the opening line of the Abstract (lines 12-13) to reflect the fact that declines are evident in multiple populations of European ground squirrels, as members of this species do not constitute a single population across their entire range.
2) Specifically stating what is meant by the "American ground squirrel" (line 131) as there is no North American ground squirrel species that goes by this as a common name, and many species exist. The authors should also provide a citation to published work containing these housing elements, including polycarbonate "shoebox" caging, feed, litter and nest material.
3) Lines 180-181 will have to be incorporated into the subsequent paragraph as one cannot have a one-sentence paragraph.
4) The second last paragraph of the Conclusions needs to be revised to refer to emergence from hibernation and not "awakening" or "waking up" as hibernation is not sleep. Further, in reporting the advantages of emerging from hibernation in better body condition for males and females, the authors must substantiate their claims with references to published primary sources.

Great work, but see additional suggestions that will hopefully allow more effective communication of your findings in the annotated copy of the PDF I've attached.
Author Response
Response letter
Authors highly appreciate the Reviewer’s very thorough evaluation of our manuscript both from scientific and language aspects. Please find our responses below.
9) The authors should report use of online random number generator to select 10 males and 10 females for the preference test. This can be achieved simply by inserting a parenthetical statement "(using an online random number generator)" after reporting "randomly selected" before the period that ends the sentence on Line 154.
Actually this is what was happened so the text was extended according to the suggestion.
My concern remains about how nest materials were "randomly" assigned to hay pockets (Line 156), as even in their reply, the authors do not describe how this was randomized, and thus leaves concern that the dispersion of nest material types offered may have influenced the preference of subjects.
Randomisation was accomplished using a random sequence generator.
I am also unclear on what a "latticed pocket" (Line 157) is, and thus suggest this be clarified for readers.
The hay racks were divided to three parts and the different nest materials we placed either to the left, middle or to the right.
Line 165 still reports pulling 20 threads at random from completed nests for analysis, without reporting how randomization was achieved. If no formal method of randomization was employed to select threads, then the authors should report that threads were selected arbitrarily or haphazardly, not randomly. Finally, the authors should explicitly that the adults employed as subjects in the nest material preference study were born into wood shavings (which I infer from their reply to my comment 9 in the previous review) as early experience has a pronounced influence on subsequent preferences.
The term haphazardly was added to the text
12) While the authors have removed the erroneous statement about dividing the 24 subjects into three groups of 8, and reported that the three feed types were presented individually in separate bowls, they have not described where the bowls were positioned, and whether placement was at all randomized or systematically controlled across trials to avoid biasing subject preference as a product of bowl placement. I also think the authors would be best served by indicating that subjects were offered a simultaneous choice of the three feed types somewhere in outlining their feed preference experiment.
Corrected in the text
13) The authors report the use of a digital scale, precise to 0.01 g to weigh feed, but should provide the model number and manufacturer of that scale, and mention in the context of weighing subject ground squirrels at the beginning and end of the study (lines 191-192) whether that same scale was used.
The requested information was added to the text
14) In response to my comment, the authors acknowledge that not all food taken from the bowls by subjects was necessarily consumed. Thus the terms "consumed" and "consumption" in describing the results of the feed preference experiment are not technically correct, and should either replaced with "taken" or the authors must define their use of the term "consumed" in their methods, to indicate that by consumed they are referring to food removed from the bowls either through immediate ingestion, or caching for subsequent ingestion by subjects.
Instead of consumed feed we use the amount of taken feed
15) While the authors have provided information to clarify and alleviate my concerns regarding the statistical analysis of their results in their reply to my comment 15 in my original review, they have not incorporated that into their revised manuscript, and should do so, so as to dispel similar uncertainty on the part of readers who may or may not make appropriate assumptions regarding the same issues I raised.
In the Statistical analysis section, we noted that Chi-square goodness of fit test was used.
Related to the ANOVA analyses we also added: The Kolmogorov–Smirnov test was used to test normality, a Levene's test was used to check the equality of standard deviations (the data corresponded to the parametric assumptions).
I also remain puzzled as to why the authors do not present, nor contrast, nest material preferences of male and female ground squirrels when they explicitly report testing 10 males and 10 females in their nest-material preference test.
We have supplemented the results section on sexes in the text.
Finally, at least some mention of inter-individual variation in preferences would be worthwhile, even if that is not tested for formally using the heterogeneity chi-square test mentioned in my original comment 15.
We confirm that testing the difference among individual nest material prefeence was not our objective. Nevertheless we confirm the it was substantially variable.
17) The authors reply to my concern regarding contamination of nest materials and feed offered with unwanted chemical cues alleviates the concern expressed in my original comment 17, but, the authors should report these precautions to readers in describing their methods in their revised manuscript.
Added to the text.
18) The authors have eliminated the redundancy between their textual account and the Figures incorporated within their results in their revised manuscript, but still must explicitly refer readers to Figures 3 and 4 (see suggested revisions in the annotated PDF to achieve this economically for Figures 3 and 4 in the Results section).
Corrected in text based on PDF
25) While the authors have done a much better job of keeping their inferences to the scope their study allows, there should still be an additional paragraph incorporated toward the end of the Discussion, acknowledging that the extent to which other attributes that varied among the feed types offered (e.g. the presence or absence of specific constituents described in column 3 of Table 1) or nutritional properties like crude ash (Table 2) contribute to the documented preference.
We regret to respond that contrary to our efforts we did not manage any study related to ground squirrels where these mentioned issues were adequately discussed.
I also suggest a few additional minor corrections, including certain grammatical revisions marked directly on the annotated PDF returned with my review, and:
1) Revising the wording of the opening line of the Abstract (lines 12-13) to reflect the fact that declines are evident in multiple populations of European ground squirrels, as members of this species do not constitute a single population across their entire range.
Corrected in the text based on PDF
2) Specifically stating what is meant by the "American ground squirrel" (line 131) as there is no North American ground squirrel species that goes by this as a common name, and many species exist. The authors should also provide a citation to published work containing these housing elements, including polycarbonate "shoebox" caging, feed, litter and nest material.
Corrected in text based on PDF
3) Lines 180-181 will have to be incorporated into the subsequent paragraph as one cannot have a one-sentence paragraph.
Corrected in text based on PDF
4) The second last paragraph of the Conclusions needs to be revised to refer to emergence from hibernation and not "awakening" or "waking up" as hibernation is not sleep. Further, in reporting the advantages of emerging from hibernation in better body condition for males and females, the authors must substantiate their claims with references to published primary sources.
Corrected in text based on PDF